# Sex-Dependent Wheel Running Effects on High Fat Diet Preference, Metabolic Outcomes, and Performance on the Barnes Maze in Rats

**DOI:** 10.3390/nu12092721

**Published:** 2020-09-05

**Authors:** Tiffany Y. Yang, Zijun Gao, Nu-Chu Liang

**Affiliations:** 1Department of Psychology, College of Liberal Arts and Sciences, University of Illinois—Urbana-Champaign, Champaign, IL 61820, USA; tyang42@illinois.edu (T.Y.Y.); zijung2@illinois.edu (Z.G.); 2Division of Nutritional Sciences, College of Agricultural, Consumer and Environmental Sciences, University of Illinois—Urbana-Champaign, Urbana, IL 61801, USA; 3Neuroscience Program, College of Liberal Arts and Sciences, University of Illinois—Urbana-Champaign, Urbana, IL 61801, USA

**Keywords:** high fat diet, wheel running, oral glucose tolerance test, Barnes maze

## Abstract

Excessive and prolonged intake of highly palatable, high fat (HF) foods contributes to the pathogenesis of obesity, metabolic syndrome, and cognitive impairment. Exercise can restore energy homeostasis and suppress HF diet preference in rats. However, it is unclear if exercise confers similar protection against the detrimental outcomes associated with a chronic HF diet preference and feeding in both sexes. We used our wheel running (WR) and two-diet choice (chow vs. HF) paradigm to investigate the efficacy of exercise in reversing HF diet-associated metabolic and cognitive dysregulation in rats, hypothesizing that beneficial effects of exercise would be more pronounced in males. All WR rats showed HF diet avoidance upon running initiation, and males, but not females, had a prolonged reduction in HF diet preference. Moreover, exercise only improved glucose tolerance and insulin profile in males. Compared to sedentary controls, all WR rats improved learning to escape on the Barnes maze. Only WR females increased errors made during subsequent reversal learning trials, indicating a sex-dependent effect of exercise on behavioral flexibility. Taken together, our results suggest that exercise is more effective at attenuating HF-associated metabolic deficits in males, and highlights the importance of developing sex-specific treatment interventions for obesity and cognitive dysfunction.

## 1. Introduction

In the United States, ~65% of adults are either overweight or obese presenting with chronic illnesses (e.g., cardiovascular disease, type 2 diabetes, hypertension, and cancer) that can be partially attributed to diet composition [1,2]. The shift in the types of food consumed and their nutritional qualities are associated with the change in environment (e.g., industrialization), including the development of agriculture, food processing, and animal husbandry [3]. Consumption of refined sugars (e.g., high-fructose corn syrup) and saturated fats has steadily increased [4,5,6] and has been termed the “Western diet,” which contains ~40% calories from both carbohydrates and fats [7]. Moreover, the modern environment favors a sedentary lifestyle and facilitates easy access to these highly processed, palatable, energy dense foods which tend to have higher glycemic loads [8] than unrefined foods and pose a threat to metabolic [9,10,11,12] and cognitive health [13].

The overconsumption of high fat (HF) food is associated with weight gain and increased abdominal adiposity, which contributes to development of peripheral metabolic dysregulation and cognitive deficits [13,14,15]. Exercise appears to be more effective than diet control at improving metabolic function [16]. People who are successful at maintaining long-term weight loss report high physical activity and a diet low in fat composition [17,18]. Rodent studies show that males consistently decrease food intake in response to exercise [19,20,21,22] whereas results are more variable in females [21,22,23]. In contrast, the majority of human studies focus on changes in food intake following acute rather than long-term exercise, and the response to exercise is highly variable in both sexes [24,25,26,27,28,29]. Although women may lose less weight than men during long-term exercise [30,31,32,33], there has been limited research to elucidate sex differences in exercise-related changes in energy intake and expenditure [34,35]. Therefore, it is unclear how closely sex differences in energy intake and macronutrient preference during exercise in humans parallel the rodent literature. Furthermore, compared with caloric restriction-mediated weight loss, exercise training-mediated weight loss led to greater decreases in visceral fat and hepatic insulin resistance in obese men and women [36]. Exercise can produce a modest improvement on peripheral glucose metabolism even without significant changes in body mass and composition. For example, short-term running exercise was able to reduce fasting glucose and portal vein free fatty acids in sucrose-fed rats without a concomitant reduction in adiposity [37]. Increasing the duration of running from four to 12 weeks in rats resulted in decreased mesenteric and subcutaneous fat in addition to increased insulin sensitivity and greater improvement in their overall metabolic profile [37]. Thus, while exercise can attenuate obesity-related insulin resistance, there may be an additive effect of exercise and adiposity loss on significantly improving peripheral insulin resistance.

Diets high in fat composition have been implicated in affecting cognitive performance in both humans [38] and rodents [39]. More specifically, chronic HF feeding has been shown to impair hippocampal-dependent spatial learning and memory in rodent models [39,40,41,42,43,44]. Given that exercise has been shown to enhance cognition [45,46,47], it may also be able to attenuate HF-induced performance deficits. Indeed, studies have found that exercise, [48,49,50,51,52] but not dietary supplementation [52], reverses HF-mediated cognitive impairment. Most studies focus on diet-induced deficits in hippocampal dysfunction. However, beyond learning and memory, HF diet can also alter prefrontal cortex (PFC)-mediated executive function, including behavioral flexibility [15,42,53,54]. Deficits in behavioral flexibility can manifest prior to the development of HF-induced insulin resistance [55]. These deficits in behavioral flexibility may result in the inability to appropriately adapt dietary choices to external environmental and internal visceral cues and consequently, promote rigid diet choices in a viscous cycle that facilitates the development of obesity [56,57]. The few studies that examined PFC-dependent cognition found that HF feeding led to deficits in behavioral flexibility in rats [15,42,53,54]. Importantly, decreased behavioral flexibility was correlated with decreased insulin sensitivity but not body weight or plasma glucose level [54]. Dietary [58,59], but not drug [60], interventions are able to attenuate these HF diet-induced cognitive impairments. Taken together, these studies suggest that HF diet and insulin resistance may interact to promote the development and maintenance of cognitive rigidity.

Sex differences in HF-induced deficits in metabolic and cognitive function may differentially affect the cognitive control of feeding behavior in males and females. The metabolic [61,62,63,64] and cognitive [65,66,67] outcomes of HF feeding appear to be more deleterious in males than females in both humans and rodents. However, obesity-related deficits, specifically in the domain of cognitive flexibility, are greater in women than men [68]. In contrast, HF-mediated deficits in behavioral flexibility are more pronounced in male rats compared to females [65,66,67]. Notably, both the human [69,70,71,72,73] and rodent [74,75,76] literature suggests that males are more responsive to the beneficial effects of exercise. Thus, exercise may more readily counteract the detrimental effects of HF-feeding in males than females. While exercise has some efficacy at counteracting HF-mediated cognitive decline in rodents [16,36,37,38,39,48,50,52], studies primarily focus on behaviors mediated by the hippocampus [48,50] rather than the PFC [49] and rarely include both sexes within the same experiment for the direct assessment of sex differences. Thus, whether exercise is able to attenuate functional dysregulation of the PFC to the same extent in both male and female rats is unclear.

To our knowledge, no study has investigated the efficacy of exercise at reversing HF-induced deficits in metabolic and executive function in rats of both sexes, which could provide insight into the development of sex-specific prevention and treatment options. Furthermore, most rodent models lack a dietary choice component so the relationships between diet preference, exercise, and metabolic function are rarely explored. To address the gap in knowledge, rats underwent our established WR and two-diet choice protocol for six weeks, a period of time which is sufficient to induce diet-induced obesity [77]. With this long-term HF diet exposure, we examined if higher HF diet preference increases susceptibility to the adverse effects of HF diet, and whether exercise can lead to similar HF-diet associated alterations in metabolic profile and cognitive performance in male and female rats. We hypothesized that chronic HF feeding would impair peripheral metabolic function and PFC-mediated behavioral flexibility in a Barnes maze [58,78] to a greater extent in female than male rats that have a preference for HF diet. Furthermore, we hypothesized that exercise would have a greater efficacy at attenuating the adverse effects HF diet exposure in males compared to females.

## 2. Materials and Methods

### 2.1. Subjects

The subjects included 24 male (250–275 g) and 24 female (150–175 g) Sprague-Dawley rats (Envigo, Indianapolis, IN, USA) that were ~7–8 weeks old upon arrival. Rats were group housed on a standard 12:12 light-dark cycle (lights on at 0700 h). During habituation, rats had ad lib access to a standard chow diet (chow; Teklad global 2018, Teklad Diets, Madison, WI, USA) and tap water. See Table 1 for details about macronutrient sources. During the experimental period, rats had diet choice between the standard, high carbohydrate chow and a novel 45% HF diet (HF; D12451, Research Diets, New Brunswick, NJ, USA). All groups were fed ad libitum.

All experimental procedures were approved by the Institutional Animal Care and Use Committee at the University of Illinois, Urbana-Champaign (Protocol #16178) and are in accordance with the Guide for the Care and Use of Laboratory Animals [79].

### 2.2. Procedures

#### 2.2.1. Wheel Running and Two-Diet Choice

Daily recording of body weight, food, and water intake occurred at 0800 h. Running activity was recorded on a computer and processed daily immediately following daily care (VitalView, Starr Life Sciences, Oakmont, PA, USA). After habituation, sedentary (Sed) rats were moved to standard individual housing cages that included cotton nesting materials as enrichment, while wheel running (WR) rats were transferred to running wheel cages (13” diameter wheel; Mini Mitter, Starr Life Sciences, Oakmont, PA, USA) with the wheel locked for a 4-day acclimatization period. After this 4-day period, a novel 45% HF diet was introduced to all rats 2 h before dark onset (1700 h) during which running wheels were simultaneously unlocked for the WR rats. The sample size, *n* = 10–11 for Sed groups and *n* = 13–14 for WR groups, was determined based on our previous studies using the same two-diet choice and wheel running paradigm [80]. The observed power with such a sample size was ≥0.80, which is the typical cutoff used for power analyses. The wheel running and two-diet choice procedures continued for ~6 weeks after which the rats were sacrificed (Figure 1). Retroperitoneal, mesenteric, and gonadal fat pads were dissected and weighed after sacrifice.

#### 2.2.2. Oral Glucose Tolerance Test (OGTT)

Two OGTTs were performed for this experiment, one at baseline and one after HF diet exposure (post-HF) to examine within-group effects of HF diet on glucose tolerance. Baseline OGTT was performed the day before the wheels were unlocked (day 0), and post-HF OGTT was performed the day of sacrifice (day 44), which was two days after the end of the Barnes maze. The day before both OGTTs, food was removed ~3 h after dark onset (2200 h) where rats would have eaten ≥60% of their daily food intake. Rats were only moderately fasted because complete overnight fasting enhances insulin-stimulated glucose utilization, and we were interested in assessing insulin action in a more physiological context [81,82]. After the baseline/fasting blood glucose (0 min) measurement and tail blood collection, rats underwent a glucose challenge where they were orally gavaged with 2 g/kg of 20% glucose dissolved in distilled water. Tail blood was collected from the same tail nick made during baseline sampling at 15, 30, 60, and 120 min time points from the time they were gavaged. Blood glucose levels were also measured at these time points using a handheld glucometer (AlphaTRAK2, Abbott, Abbott Park, IL, USA). Tail blood was centrifuged at 870× *g* for 15 min at 4 °C. For each sampling time point, ~25 µL of plasma was collected and stored at −80 °C until the samples were processed for plasma insulin concentrations using the Rat Ultrasensitive Insulin ELISA (ALPCO, Salem, NH, USA) according to manufacturers’ protocol.

Blood glucose and plasma insulin data from the baseline and post-HF OGTT was used to assess insulin sensitivity [83] with the following two methods: (1) Hepatic insulin resistance was calculated using the homeostasis model assessment method for insulin resistance (HOMA-IR) model [84] and (2) peripheral insulin resistance was calculated using Gutt’s index of insulin sensitivity (ISI_0,120_) [85].
$$\mathrm{HOMA}-\mathrm{IR}\;=\;\frac{\mathrm{fasting}\;\mathrm{insulin}\;\left(\frac{\mu U}{\mathrm{ml}}\right)\times \mathrm{fasting}\;\mathrm{glucose}\;\left(\frac{\mathrm{mmol}}{L}\right)}{22.5}$$
(1)$$\begin{array}{l} ISI_{0,120}\\ =\frac{\mathrm{glucose}\;\mathrm{load}\;\left(\mathrm{mg}\right)+\left({\mathrm{glucose}}_{0\;\min}-{\mathrm{glucose}}_{120\;\min}\;\left(\frac{\mathrm{mg}}{L}\right)\right)\times 0.19\times \mathrm{body}\;\mathrm{weight}\;\left(\mathrm{kg}\right)}{120\times \log\left(\frac{{\mathrm{insulin}}_{0\;\min}+{\mathrm{insulin}}_{120\;\min}\left(\frac{\mathrm{mU}}{L}\right)}{2}\right)+\left(\frac{{\mathrm{glucose}}_{0\;\min}+{\mathrm{glucose}}_{120\;\min}\left(\frac{\mathrm{mmol}}{L}\right)}{2}\right)} \end{array}$$

#### 2.2.3. Barnes Maze

During the last week of the WR and two-diet choice period, all rats were trained on the Barnes maze starting 2.5 h after light onset (0930 h). A concealed overhead-mounted camera aimed directly at the center of the maze was used to film each trial, which was operated using a computer from the adjacent room. The Barnes maze was 99 cm high and 122 cm in diameter, with 20 evenly spaced holes that were 10 cm in diameter and 2 cm away from the edge. The apparatus was mounted on a rotatable wooden support system that allowed the maze to be rotated 360°. The escape box (30 × 12.5 × 14.5 cm) was mounted underneath one hole with a 20° incline ramp. The location of five visuospatial cues was held constant during training and reversal learning. For both training and reversal learning trials, a trail ended when the rat entered the escape box or after the allotted time had elapsed. The maze and escape box were cleaned using a non-alcohol-based coverage spray between each rat to eliminate odor cues.

After daily care, rats were single caged in standard tubs and moved to the room adjacent to the testing room for at least 1 h of habituation prior to testing. There were 4 trials/day during training (total of 16 trials) with 4 different starting locations (Figure 2). Rats were placed on the edge of the maze facing the wall/away from the center of the maze. All rats finished a trial at the first starting location with an inter-trial interval of 30 min before being tested at the second starting location. In other words, all rats completed a trial from the same starting location before the next round of trials began. The order of the starting locations remained the same for each training day. Rats were given 90 s to find the escape box, and if a rat failed to find the escape box within the allotted time, they were gently guided into the box, which was then covered. Rats were allowed to remain in the escape box for 15 s before being returned to their home cage.

On the fifth day of the Barnes maze, rats were placed at the center of the maze facing away from the escape box for a probe trial to ensure they learned the task. The procedures were the same from testing. After the probe trial, the escape box was rotated 180°, but none of the visuospatial cues were moved. For the 3 reversal learning trials with 30 min inter-trial intervals, rats were given 150 s to locate the new location of the escape box and were gently guided in if they failed to find the escape box and allowed to remain in the box for 15 s.

### 2.3. Statistical Analysis

Statistical analyses were performed using Statistica 13.3 (TIBCO, Palo Alto, CA, USA). Data are presented as the mean ± standard error of the mean (SEM). Post hoc Fisher’s LSD tests were performed when significant main effects or interactions were identified.

For the WR and two-diet choice portion, raw data included weekly averages of body weight, energy intake, and running activity. These measures were analyzed separately using 3-way mixed model ANOVAs with sex (male vs. female) and exercise (Sed vs. WR) as between-subject factors and time (6 weekly averages) as the within-subject factor. Diet choice was analyzed using a 4-way mixed model ANOVA with sex and exercise as the between-subject factors and diet (chow vs. HF) and time (6 weekly averages) as the within-subject factors.

Raw data from OGTT included plasma glucose and insulin measurements. Baseline OGTT and post-HF OGTT results were analyzed separately using a 3-way mixed model ANOVAs with sex and exercise as between-subject factors, and time (0, 15, 30, 60, and 120 min) as the within-subjects factor. Glucose and insulin area under curve (AUC) results were analyzed separately using a 2-way mixed model ANOVA with sex and exercise as the between-subject factors and time (baseline vs. post-HF) as the within-subjects factor. Two measures of insulin sensitivity were first calculated using HOMA-IR and ISI_0,120_ and then analyzed separately using a 2-way mixed model ANOVA with sex and exercise as the between-subject factors and time (baseline vs. post-HF) as the within-subject factor. A 2-way ANOVA with sex and exercise as the between-subject factors was performed to analyze trunk plasma insulin levels at the termination of the experiment. In addition, separate correlation analyses were performed to determine if there was an association between glucose AUC during OGTT, insulin AUC during OGTT, and trunk plasma insulin levels with average HF diet preference ratio.

For the Barnes maze, the training days were manually scored for latency to enter the escape box and errors. The test day was manually scored for the same measures. An error was counted each time the rat checked a hole other than the one leading to the escape box by poking its nose into the hole. At least two individuals video scored the training and testing portion of the Barnes maze for all rats. Training and testing data were analyzed using a 3-way mixed model ANOVA with sex and exercise as between-subject factors and trial (daily average) as the within-subject factor.

## 3. Results

### 3.1. Wheel Running and Two-Diet Choice

Sedentary rats decreased HF diet intake and increased chow intake across time whereas WR rats expressed the opposite diet choice pattern (time × diet × exercise F (5,220) = 37.35, *p* < 0.001; Figure 3A–D). Upon initial access to the two-diet choice, Sed rats showed extreme preference for the HF diet whereas WR rats avoided it. Subsequently, these opposite diet choice patterns were reflected as Sed rats decreased and WR rats increased HF diet preference over time (time × exercise F (5,220) = 42.28, *p* < 0.001; Figure 3E,F). Furthermore, HF diet preference did not appear to be influenced by sex, i.e., all WR females and 12 out of 14 WR males reversed HF diet avoidance (sex F (1,44) = 1.63, *p* > 0.20). However, when examining the average ratios of HF diet preference across the duration of two-diet choice, nine out of 14 WR males had a HF diet preference ratio < 0.5, which indicates that they preferred HF to chow diet for less than half of the six weeks choice period. In addition, WR females reversed HF diet avoidance earlier than males and expressed greater preference for HF diet (time × sex × exercise F (5,220) = 3.92, *p* < 0.05).

### 3.2. Running Activity and Energy Intake

Females ran more than males, and both sexes showed an inverted-U trend in running activity where running activity peaked and then decreased to baseline levels (time × sex F (5,125) = 9.73, *p* < 0.001; Figure 4A). There were sex-specific adaptations in energy intake to exercise (sex × exercise F (1,440) = 28.26, *p* < 0.001) across time (time × sex × exercise F (5,220) = 4.97, *p* < 0.001; Figure 4B). WR led to an initial decrease in total energy intake in males after which they increased food intake, but total energy intake was not different among Sed and WR males (post hoc *p* > 0.15). Conversely, female WR rats increased their total energy intake earlier than males and had significantly higher energy intake than their Sed counterparts (post hoc *p* < 0.001).

### 3.3. Body Weight and Adiposity

Exercise-mediated changes in total daily energy intake resulted in suppressed body weight gain in both males and females (exercise F (1,44) = 29.16, *p* < 0.001). Although exercise suppressed weight gain in females (Figure 5A), the difference in percent weight gain between Sed and WR rats at the end of the experiment was 10% in males and only 2% in females. The difference in body weight was reflected in fat composition. Exercise led to decreased retroperitoneal and mesenteric fat (exercise F (1,44) = 11.62 and 10.47, respectively, both *p* < 0.01) in both sexes (sex x exercise F (1,44) = 1.26 and 3.68, respectively, both *p* > 0.06; Figure 5B). Although the sex x exercise interaction did not reach statistical significance in the mesenteric fat pad (*p* = 0.061), post hoc tests indicate that the effect of exercise was driven by males. For the gonadal fat pad, there was a sex x exercise interaction where exercise resulted in a loss of gonadal fat only in males (F (1,44) = 8.16, *p* < 0.01; post hoc male Sed vs. WR *p* < 0.01 and female Sed vs. WR *p* > 0.49).

### 3.4. Oral Glucose Tolerance Test (OGTT)

At chow only baseline (BL), there were no group differences in glucose clearance following an oral glucose challenge (time × sex × exercise F (4,176) = 1.78, *p* > 0.32; Figure 6A). Following six weeks of chronic HF feeding, females had higher fasting blood glucose levels than males (time × sex F (4,172) = 5.26, *p* < 0.001; post hoc 0 min males vs. females 103.33 vs. 116.61 mg/dL, *p* < 0.05; Figure 6B). However, blood glucose levels returned to pre-glucose challenge levels in females, but not males (post hoc 0 min vs. 120 min female *p* > 0.32 and male *p* < 0.001). Analysis of area under curve (AUC) of blood glucose during the baseline and post-HF diet OGTT indicated that exercise decreased glucose AUC in males, but not females (time × sex × exercise F (1,43) = 6.81, *p* < 0.05; post hoc male WR BL vs. HF *p* < 0.01 and female WR BL vs. HF *p* > 0.12; Figure 6C).

Prior to the WR and two-diet choice experimental period, there were no baseline differences in insulin levels during an OGTT (time × sex × exercise F (4,156) = 0.85, *p* > 0.49; Figure 6D). During the post-HF diet OGTT, there was a trend for exercise to decrease plasma insulin that appeared to be driven by males (sex × exercise F (1,41) = 2.87, *p* = 0.09; Figure 6E). A one-way ANOVA revealed a group difference at 0 min, such that exercise resulted in lower plasma insulin levels only in males following long-term HF diet exposure (group F (1,42) = 4.05, *p* < 0.05; post hoc male Sed vs. WR *p* < 0.01 and female Sed vs. WR *p* > 0.13). Following an oral glucose challenge, there was no exercise effect in plasma insulin levels across different time points (time × exercise F (4,164) = 1.61, *p* > 0.17). On average, males had higher insulin AUC levels than females (sex F (1,35) = 4.41, *p* < 0.05; Figure 6F). There was also a time × sex × exercise effect (F (1,35) = 5.29, *p* < 0.05) where exercise decreased insulin AUC in males but not females. Chronic HF feeding resulted in higher insulin AUC than baseline levels in Sed males (post hoc M Sed BL vs. HF *p* < 0.001), and exercise suppressed this increase (post hoc M WR BL vs. HF *p* > 0.06). Male WR rats had lower insulin AUC post-HF diet exposure than their Sed counterparts (post hoc HF M Sed vs. M WR *p* < 0.001). In females, however, exercise did not suppress the amount of plasma insulin needed to clear the same dose of glucose (post hoc F Sed and F WR BL vs. HF both *p* < 0.01 and HF F Sed vs. F WR *p* > 0.98).

Male Sed rats had impaired hepatic insulin sensitivity following HF feeding (time × sex × exercise F (1,40) = 7.73, *p* < 0.01; post hoc M Sed BL vs. post-HF *p* < 0.0001) as evidenced by increased HOMA-IR index (Table 2). Exercise protected against this detrimental metabolic effect of HF diet in male WR rats by attenuating increases in insulin resistance (post hoc WR BL vs. post-HF *p* > 0.18 and post-HF Sed vs. WR, *p* < 0.01). In females, both Sed and WR rats had evidence of insulin resistance after long-term HF diet preference (post hoc Sed and WR BL vs. HF, both *p* < 0.05) and exercise did not have the same protective effect as seen in males (post-hoc post-HF F Sed vs. F WR *p* > 0.08). Peripheral insulin sensitivity was analyzed using ISI_0,120_ and sex differences in the protective effect of exercise failed to reach statistical significance (time × sex × exercise F (1,43) = 2.29, *p* > 0.13). However, a priori t-tests revealed that post-HF feeding, male Sed rats had lower ISI_0,120_ than their WR counterparts, indicating reduced insulin sensitivity in the Sed but not WR group (t(10,14) = −2.17, *p* < 0.05). This difference between Sed and WR groups was absent in females (t(10,13) = −0.21, *p* > 0.83).

Analysis of trunk plasma insulin after 6 weeks of HF feeding revealed that WR females had higher plasma insulin than their Sed counterparts (M Sed 1.04 ± 0.10, M WR 0.54 ± 0.05, F Sed 0.44 ± 0.06, and F WR 0.69 ± 0.06 ng/mL; sex × exercise F (1,44) = 31.40, *p* < 0.001; post hoc female Sed vs. WR *p* < 0.05) whereas the opposite pattern was observed in Sed and WR males (post hoc *p* < 0.001). Moreover, a regression analysis revealed a positive correlation between average ratios of HF diet preference and plasma insulin levels at sacrifice in males (F (1,22) = 7.72, R = 0.51, *p* < 0.01; Figure 7A) but not females (F (1,22) = 0.95, R = 0.01 *p* > 0.94; Figure 7B). No such correlation was found between average ratios of HF diet preference and either glucose or insulin AUC.

### 3.5. Barnes Maze

During training, there was a sex difference in latency whereby males were slower to locate the escape box than females (sex and trial × sex F (1,43) = 66.58 and F (3,129) = 7.59, respectively, both *p* < 0.001; Figure 8A,C). In rats of both sexes, exercise led to decreased latency to locate the escape box (trial × exercise and trial × sex × exercise F (3,129) = 4.23 and 2.00, *p* < 0.01 and *p* > 0.11, respectively). Although an effect of trial by exercise interaction on errors committed across training days reached statistical significance (F (3,129) = 3.37, *p* < 0.05), post hoc tests revealed no specific group differences on any given day. Thus, exercise slightly reduced the numbers of errors made during learning on the Barnes maze in both sexes (trial × sex × exercise F (3,129) = 0.65, *p* > 0.58; Figure 8B,D). On average, male rats made more errors than females during training (sex F (1,43) = 5.25, *p* < 0.05).

A factorial ANOVA on the probe trial revealed that there was no effect of sex or exercise on task acquisition in regards to latency (sex × exercise F (1,43) = 1.80, *p* > 0.18) and errors made (F (1,43) = 1.67, *p* > 0.20). There was also no effect of sex or exercise on errors made or latency to locate the escape box during reversal learning (sex × exercise F (1,43) = 0.79 and 0.15, respectively, both *p* > 0.37). However, when the percent increase in errors made between the probe and reversal trials was analyzed using a factorial ANOVA, there was a sex × exercise effect (F (1,34) = 6.48, *p* < 0.05). Post hoc analyses revealed that female WR rats increased more errors than their Sed counterparts (post hoc *p* < 0.05) whereas this effect was not seen in male rats (post hoc male Sed vs. WR *p* > 0.10).

Both male and female rats used a non-spatial serial search strategy (Appendix A), e.g., a clockwise or counterclockwise sequential search, rather than a direct search strategy where the rats utilize spatial cues, e.g., signs in the testing room to find the escape box. Despite not using the visual cues, the decreased latency and errors indicated that all rats learned the task (Appendix A). While Sprague-Dawley rats have poor visual acuity biasing them towards using a serial search strategy [78], visual acuity has not been correlated with deficits in learning and memory issues in mice tested on the Barnes maze [86]. Moreover, our results for latency and errors made are comparable to what has been reported in the literature [78,87].

## 4. Discussion

Currently, it is unclear whether exercise has a similar efficacy at reversing the adverse metabolic and cognitive effects of HF preference and intake in rats of both sexes. To address this, we used a long-term two-diet choice and WR model to examine the relationship between preference for HF diet and the detrimental metabolic and cognitive outcomes associated with chronic HF feeding, and whether exercise has the ability to attenuate these negative effects. We found that both male and female WR rats recovered from their initial running-induced HF diet avoidance and increased both HF diet intake and preference across time (Figure 3C,D). Exercise had a protective effect in males, but not females, on HF-mediated weight gain and adiposity (Figure 5) and metabolic dysfunction (Figure 6C,F). Moreover, the positive association between preference for HF diet and trunk plasma insulin was only seen in males (Figure 7A). Wheel running rats had significantly faster escape latencies and made slightly fewer errors than their sedentary counterparts in both sexes across training days, suggesting improved learning and memory through exercise (Figure 8). Although our interpretation of behavioral flexibility was limited due to the utilization of a non-spatial search strategy by the rats (Appendix A), female WR rats were the only group to increase the number of errors made between the probe and reversal learning trials compared to their Sed controls. Taken together, our results suggest that exercise-mediated changes in HF diet preference lead to sex-specific effects in regards to the protective effect of exercise on both peripheral metabolic function and cognitive performance.

The opposite diet choice patterns observed among Sed and WR rats (Figure 3A–D) may be due to the increased metabolic requirement from exercise to maintain energy balance where the HF diet is a more efficient fuel source than the standard chow diet, which is higher in carbohydrates than fats [88,89]. Indeed, human studies have shown that there is a crossover effect during which the ratio of lipolysis to carbohydrate oxidation during submaximal and endurance exercise increases [90,91,92]. While this crossover effect is influenced by exercise duration and intensity, it may partially contribute to differences in macronutrient preference among sedentary and physically active individuals. Our previous short-term studies with the same paradigm of wheel running and two-diet choice revealed that the majority of male rats express persistent HF diet avoidance whereas the majority of females reverse HF diet avoidance [80,93,94,95]. These results are consistent with the report that estradiol enhances lipid metabolism during exercise in rats [96]. Furthermore, results with respiratory exchange ratio as a measure of substrate utilization from human studies using indirect calorimetry suggest that compared to men, women utilize more fat as the fuel source as a result of long-term exercise [90,91,97,98]. A direct assessment of fuel oxidation will be necessary to support our hypothesis that sex differences in substrate utilization may contribute to differences in running-associated macronutrient preference. When the choice duration was extended, both male and female WR groups preferred the HF diet by the end of the six-week period (Figure 3E,F). It is unclear whether this increase in fat preference would occur in humans if the behavior can be examined without the influence of the cognitive component of making healthier food choices in subjects who incorporate regular exercise as a lifestyle. Nevertheless, carbohydrate metabolism is positively correlated with exercise intensity in humans [99] whereas fat oxidation is more likely to occur during low intensity exercise, especially when prolonged [100]. Thus, this shift in fat preference may be a compensatory result of increased energy requirement from long-term aerobic exercise where carbohydrates are no longer the most efficient fuel substrate. The addition of groups undergoing different types of exercise (e.g., strength, treadmill, swimming, etc.) for different lengths of time would provide additional evidence for the effect of exercise on energy intake and macronutrient preference if results are consistent.

Although both male and female WR rats reversed their initial avoidance for HF diet, there was a sex difference in which the reversal of HF diet avoidance occurred with females reversing earlier than males. One potential explanation for this sex difference is that females are more prone to hedonic and binge eating than males [101,102] and their feeding behavior appears to be driven by palatability rather than physiological hunger or metabolic state [103,104]. Both of these factors could act together and exacerbate the development of obesity [56,57], which is more prevalent in females [105]. In addition, females have higher reward sensitivity than males [106,107] which may predict decreased restraint of fat intake [108]. HF diet is highly palatable and can stimulate eating in the absence of hunger by acting on the reward system [104], potentially leading to overeating. Although WR is naturally rewarding for rodents [109], it may not be a sufficient substitute for the reinforcing effects of the palatable HF diet for females [110]. In support of this, studies have shown that male rats are more responsive to the reinforcing effects of voluntary WR to attenuate seeking of drugs of abuse [111,112]. The high sensitivity to WR reinforcement could facilitate males’ ability to maintain lower preference for HF diet for a longer duration than females that are more sensitive to the reinforcing effects of diet palatability [110].

Sex differences in adapting to the increased energy requirement of exercise led to differential efficacies for exercise to attenuate HF-mediated insults on metabolic function in male and female rats. Consistent with the literature [21,23,113], females ran more than males (Figure 4A) and compensated for the increased energy expenditure by increasing total energy intake earlier than males [22,114]. This may have led to the limited effect of exercise on suppressing body weight and adiposity [19,20,21,22] and attenuating HF mediated metabolic dysregulation in females (Figure 6). Here, we report that exercise suppressed HF diet preference, total energy intake, body weight, and adiposity, and improved glucose metabolism to a greater degree in males than females. This is consistent with the consensus in the literature stating that males are more responsive to the beneficial effects of exercise, resulting in improved glucose tolerance and insulin sensitivity [74,75,76]. Thus, it appears that exercise has a protective effect on insulin sensitivity in males despite increased HF diet intake [69,70,73,115,116,117,118,119,120]. We also found a positive association between HF diet preference and insulin levels in males but not females (Figure 7). The greater protective effect of exercise on peripheral metabolic function in males may be mediated by two effects acting in concert: (1) a slower compensatory response to the increased energy expenditure from exercise [114] and (2) maintenance of a lower preference for HF diet for a longer duration of time compared to females.

Our results suggest that that without a concurrent decrease in body weight, adiposity, and HF diet preference, exercise has a limited effect on significantly improving peripheral insulin resistance during chronic access to HF diet [16,36,37]. Female rats had increased HOMA-IR (Table 2) above the 2.60 cutoff [121] for evidence of hepatic insulin resistance after long-term HF feeding regardless of the opportunity to exercise. In contrast, exercise appeared to protect against the development of insulin resistance in males. The results of ISI_0,120_ also suggest that male WR rats were the only group that maintained insulin sensitivity after six weeks of exposure to HF diet. Although higher HF diet and total energy intake in WR females may play a role in these observed sex differences and additional pair-fed groups will be needed to assess such possibility in future studies, our results reveal that exercise produces more protective effects against insulin resistance in males than females by reducing HF diet preference and consumption. Consequently, an optimal treatment for weight loss and improving insulin sensitivity for females would be a combination of diet and exercise [122].

The improved performance during training on the Barnes maze in WR rats of both sexes relative to their Sed counterparts suggests that exercise can be protective against insults to cognitive behavior from chronic HF diet consumption independent of sex (Figure 8). This aligns with rodent literature linking HF feeding to deficits in cognitive behaviors [15,42,53,54], which exercise can reverse [44,48,49,50,51,52]. In contrast to reports that male and female rats performed similarly on the Barnes maze [123,124], we found that females had faster latencies than males to locate the escape box. Rather than enhanced learning, this effect could be a result of hormonal/estrous status or higher general locomotor activity exhibited by females [125,126]. Although our results conflict with previous literature reporting either a male advantage or no difference in performance between males and females, sex differences have not been consistently reported in regards to learning visuospatial tasks [86,127,128,129]. Moreover, task performance is influenced by a variety of factors including task design, species, strain, hormonal status, stress, and age [130]. To our knowledge, there have been a limited number of studies that investigate sex differences in cognitive behavior using the Barnes maze in rats [127,128,131] with the Morris water maze being the more popular task [130]. More standardized research is necessary to draw firm conclusions on subtle sex differences in cognitive and spatial ability given how the Barnes maze is sensitive to a variety of conditions that may influence task performance and subsequent interpretation of the data.

Despite similar search strategies and learning outcomes, there was evidence of sex-specific effects of running on reversal learning on the Barnes maze. The non-spatial serial search strategy (Appendix A) is not uncommon in Sprague-Dawley rats [128] that have lower visual acuity than other strains (e.g., Long-Evans and Wistar) [128,132] where using a serial search strategy may be more efficient. Notably, the increase in errors made from the probe to reversal learning trials was higher in WR females compared to their Sed counterparts, whereas there was no such group difference in males. This sex-specific effect may be interpreted as WR females having the worst behavioral flexibility among all groups, which conflicts with the finding that PFC-mediated deficits in behavioral flexibility are more pronounced in males than females following HF feeding [65,66,67]. However, our interpretation of the data is limited such that (1) rats did not appear to use spatial cues to locate the escape box and (2) learning without utilizing spatial cues may render weak task acquisition and as such, poses a confound for the analysis of potentially increased errors during the reversal learning trials. Thus, an alternate interpretation to our result that only female WR rats significantly increased their errors during reversal learning may be that this was the only group that acquired the original task and could exhibit deficits in behavioral flexibility. Nevertheless, while adjustments can be made to promote the use of spatial cues (e.g., moving cues closer to the maze, adding an aversive stimulus/bright light, etc.), the Barnes maze may not be the most optimal behavioral task to assess subtle deficits in cognitive behavior. In humans, obesity generally results in mild rather than severe cognitive impairment [133,134]. Therefore, a more sensitive behavioral task may allow us to uncover cognitive deficits more readily than the Barnes maze. Cognition is a complex and multi-faceted construct. Thus, while we found evidence for differences in cognitive performance, future studies should include a battery of behavioral tests to tap into different aspects of cognitive behavior that may be adversely influenced by HF diet. Importantly, prolonged HF diet intake and preference may impair specific, rather than global, domains of cognitive function in a sex-specific fashion. Impairments in one domain may be more influential in the regulation of feeding behavior and lead to worse outcomes depending on sex.

Caution should be taken when interpreting the results due to inherent limitations resulting from the complexity of the study design. Different fat sources and compositions (e.g., polyunsaturated, monounsaturated) may differentially affect aspects of metabolism and cognition. Future studies should utilize diets matched in macronutrient sources as much as possible to limit the confounding effect of differences in raw materials. To our knowledge, no study has assessed changes in insulin sensitivity following HF diet and voluntary exercise in rats of both sexes. The addition of a naïve chow-fed control group is necessary to make between-group comparisons to strengthen the argument for the protective effect of exercise against HF diet. Exercise is more likely to have a greater beneficial effect in males, but the addition of a WR group maintained on only HF diet is necessary to make this conclusion given the shift in HF diet preference across time and the highly variable HF diet intake between subjects and sexes. Here, we focused on voluntary exercise; however, the effect of exercise may shift depending on exercise conditions (forced vs. voluntary, strength vs. endurance, acute vs. chronic, etc.). An extensive investigation is necessary before results can be directly translated to humans. Nevertheless, our results lend support to our hypotheses regarding the protective effect of exercise against the detrimental outcomes of HF feeding and inform the development of more optimized designs for future studies.

## 5. Conclusions

We examined sex differences in exercise-mediated changes in diet choice and the degree to which exercise can reverse the metabolic dysregulation and improve cognitive performance associated with long-term HF feeding. The protective effect of exercise on suppressing HF diet preference and HF-mediated insults to peripheral metabolism was specific to males whereas exercise similarly enhanced learning on the Barnes maze in both males and females. Intriguingly, despite less improvement on their metabolic profile, female WR rats still benefited from the exercise and showed improved performance on the Barnes maze. This finding suggests that cardio-based exercise can potentially exert differential effects on metabolic and cognitive function. Taken together, these results suggest that the adverse metabolic effects of chronic HF feeding and preference are especially detrimental to females, but exercise remains a good intervention option for both males and females to prevent cognitive decline resulting from poor dietary choices.

## Figures and Tables

**Figure 1 nutrients-12-02721-f001:**
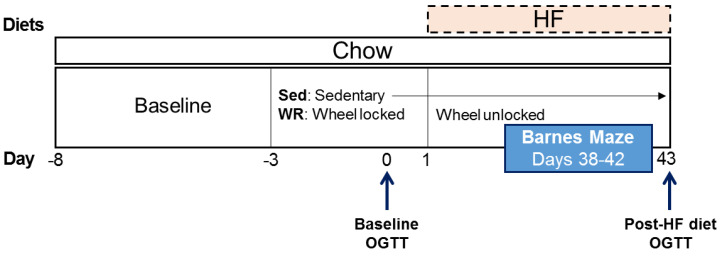
Experimental timeline. Sed: sedentary, WR: wheel running; HF: high fat; OGTT: oral glucose tolerance test.

**Figure 2 nutrients-12-02721-f002:**
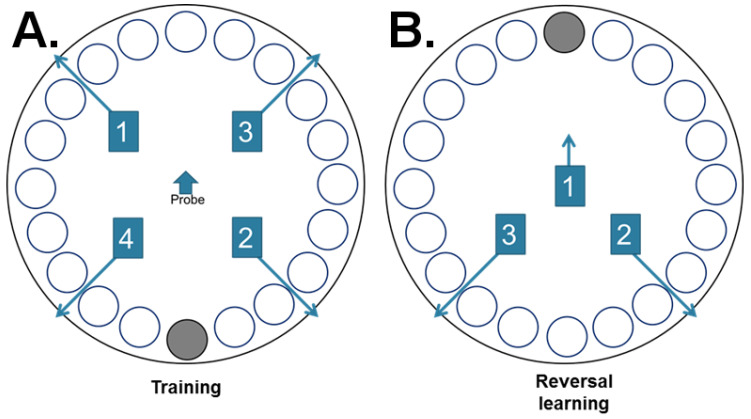
Barnes maze. Starting locations on the Barnes maze for training and reversal learning trials. (**A**) Rats underwent 4 trails/day during training. (**B**) On the testing day, rats went through a probe trial to assess task acquisition and 3 reversal learning trials to assess behavioral flexibility.

**Figure 3 nutrients-12-02721-f003:**
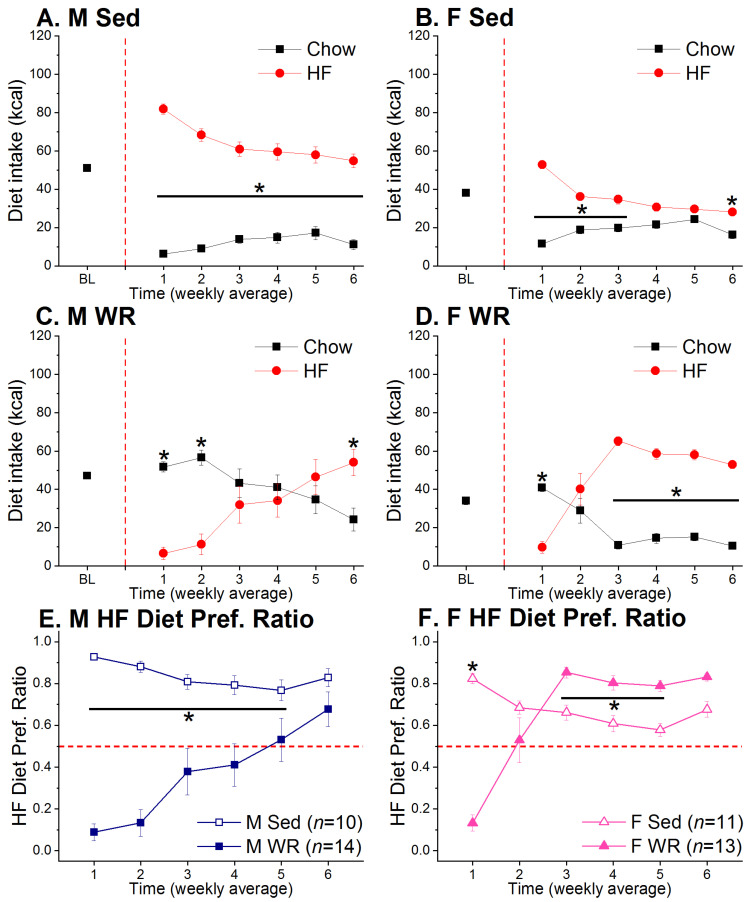
Diet choice and HF diet preference ratios in males (M, left) and females (F, right). WR rats expressed opposite diet choice patterns from their Sed controls. The vertical and horizontal dashed red lines denote the start of the two-diet choice and WR period and preference for HF diet where any value greater than 0.5 indicates a preference for HF diet, respectively. (**A**,**B**) Sedentary male rats maintained higher intake of HF than chow diet throughout the experiment whereas there was a two-week period in which Sed females did not show a preference for either diet. * chow vs. HF, *p* < 0.05. (**C**,**D**) Both male and female WR rats increased HF diet intake across time. The reversal of HF diet avoidance occurred earlier in females than males. * chow vs. HF, *p* < 0.05. (**E**,**F**) HF diet preference went in opposite directions among Sed and WR rats in both sexes. * Sed vs. WR, *p* < 0.05.

**Figure 4 nutrients-12-02721-f004:**
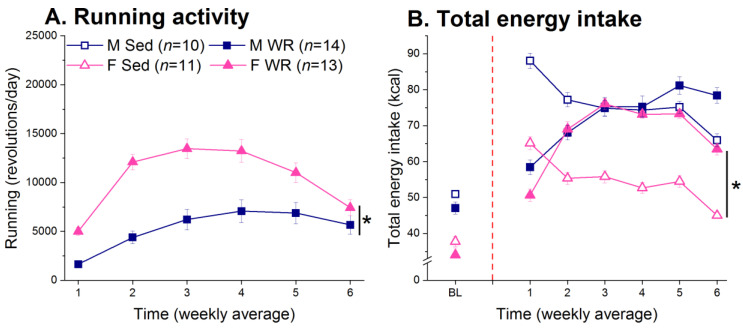
Running activity and total energy intake. (**A**) Female rats ran more than males, and both sexes showed and inverted-U trend in running activity. * Male vs. female, *p* < 0.05. (**B**) Female, but not male, running rats had higher energy intake than their Sed counterparts. * Sed vs. WR, *p* < 0.05.

**Figure 5 nutrients-12-02721-f005:**
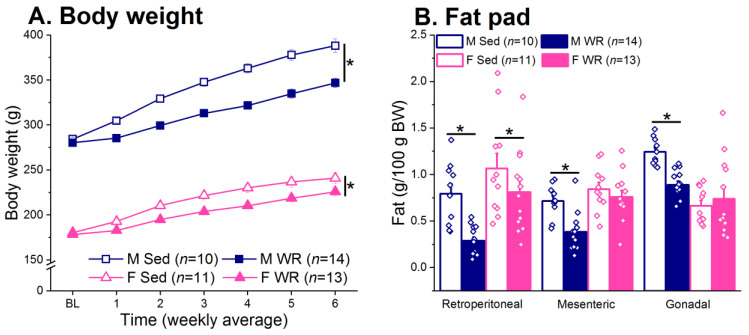
Body weight and fat composition. (**A**) Exercise suppressed body weight in both sexes; however, this effect was more pronounced in males. * Sed vs. WR, *p* < 0.05. (**B**) Exercise led to decreased retroperitoneal, mesenteric, and gonadal fat in males and decreased only retroperitoneal adiposity in females. * Sed vs. WR, *p* < 0.05.

**Figure 6 nutrients-12-02721-f006:**
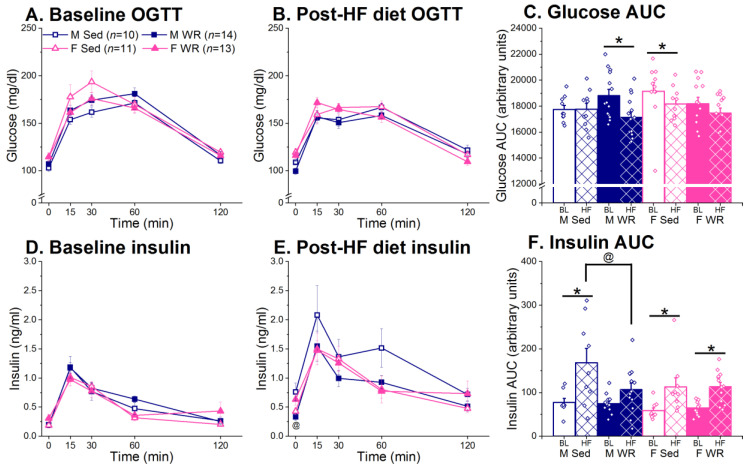
Blood glucose (top row) and plasma insulin (bottom row) results from an OGTT at baseline (BL) and post-HF diet exposure. (**A**) There were no group differences in blood glucose at BL. (**B**) Blood glucose levels following an oral glucose challenge returned to fasting levels faster in females than males. (**C**) Decreased glucose clearance, indicated by AUC (area under curve), from BL occurred in male WR and female Sed rats. * BL vs. HF, *p* < 0.05. (**D**) There were no group differences in plasma insulin at BL. (**E**) Following HF diet exposure, exercise decreased fasting plasma insulin levels in males but not females. @ M Sed vs. WR, *p* < 0.05. (**F**) Male WR rats had lower insulin AUC than their Sed counterparts. In contrast, both Sed and WR females had higher insulin AUC post-HF exposure than at chow BL. * BL vs. HF, *p* < 0.05, @ HF M Sed vs. WR, *p* < 0.05.

**Figure 7 nutrients-12-02721-f007:**
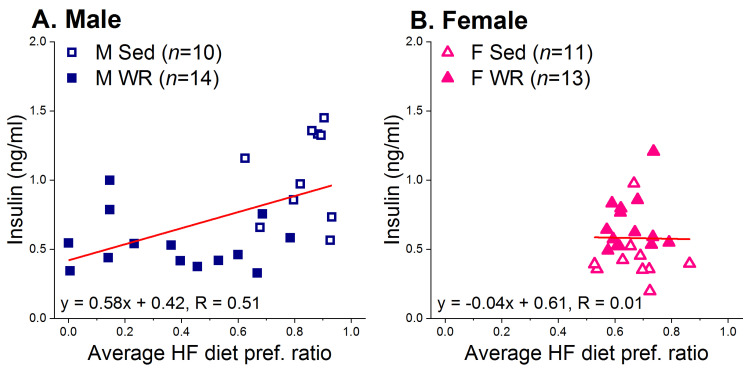
Correlation between trunk plasma insulin levels at the end of the experiment and the average ratios of HF diet preference. (**A**) There was a moderate, positive correlation between HF diet preference and trunk plasma insulin in males. (**B**) There was no relationship between HF diet preference and trunk plasma insulin levels in females.

**Figure 8 nutrients-12-02721-f008:**
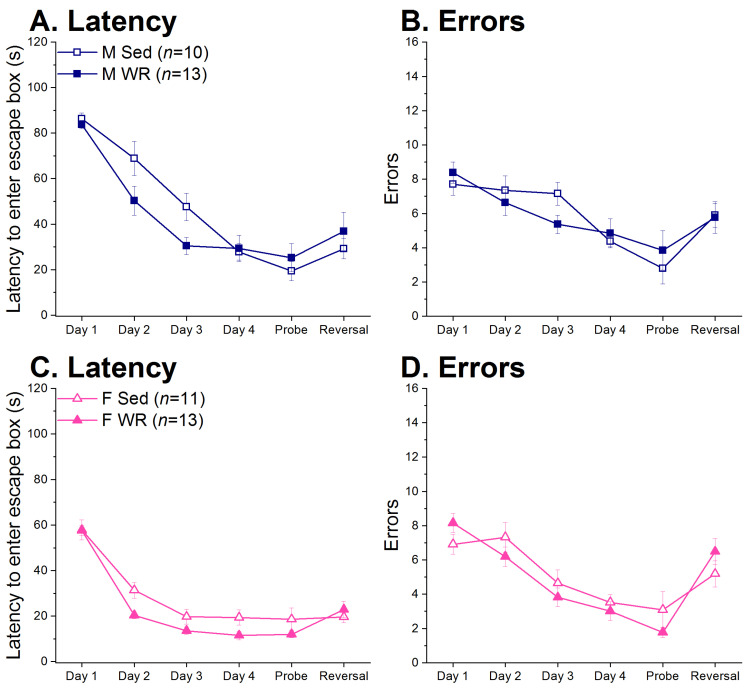
Barnes maze results for males (M, top) and females (F, bottom). (**A**,**C**) Both male and female rats decreased latency to locate the escape box during training. On average, females had shorter latencies to find the escape box than males across training days. (**B**,**D**) All groups committed fewer errors when searching for the escape box across training days.

**Table 1 nutrients-12-02721-t001:** Diet composition.

Macronutrient	Description	Unit	Teklad 2018 (3.1 kcal/g)	45% HF Diet (4.74 kcal/g)
Protein	Total	% kcal	24	20
Carbohydrate	Sucrose	% kcal	-	17
	Other carbohydrates	% kcal	58	18
Fat	Total	% kcal	18	45
	Saturated fats	% of total fat (wt)	0.9	31.4
	Monounsaturated fats	% of total fat (wt)	1.3	35.5
	Polyunsaturated fats	% of total fat (wt)	3.4	33.1

**Table 2 nutrients-12-02721-t002:** Hepatic and peripheral indices of insulin sensitivity. Exercise protected against the development of insulin resistance by HF diet to a greater extent in males than females.

	Insulin Sensitivity	HOMA-IR		ISI_0,120_	
Group		Baseline	Post-HF	Baseline	Post-HF
Male Sed		1.22 ± 0.31 *	5.01 ± 0.99 ^	0.79 ± 0.04 *	0.63 ± 0.02 ^
Male WR		1.34 ± 0.30	2.15 ± 0.45	0.74 ± 0.03	0.71 ± 0.03
Female Sed		1.43 ± 0.38 *	3.15 ± 0.47	0.71 ±0.03	0.64 ± 0.02
Female WR		2.25 ± 0.49 *	4.63 ± 0.81	0.70 ± 0.03	0.65 ± 0.02

HOMA-IR: Homeostatic assessment of insulin resistance; ISI_0,120_: Gutt’s insulin sensitivity index (mg × L^2^ × mmol^−1^ × mU^−1^); *: Baseline vs. Post-HF, *p* < 0.05; ^: Sed vs. WR, *p* < 0.05. Data are represented as the mean ± SEM.

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
