# Peer review of "Sex-Dependent Wheel Running Effects on High Fat Diet Preference, Metabolic Outcomes, and Performance on the Barnes Maze in Rats"

_nutrients, 2020, doi:10.3390/nu12092721_

Round 1

Reviewer 1 Report

Nutrients – 892060

‘Sex-dependent wheel-running effects on high fat diet preference, metabolic outcomes, and performance on the Barnes maze in rats’ by Yang, Gao and Liang

The manuscript suggests to investigate the influence of high fat diet on the capability of exercise to suppress negative diet effects on body characteristics, metabolism and cognitive function in rats. While the topic is not exhaustively investigated in the literature, I have major concerns about the design, analysis and interpretation of the study.

Major concerns:

Design – the authors used a free choice paradigm between high fat food and chow (CHO rich) over a period of 6 weeks. The design is identical to their formerly published work (Yang et al. 2019) addressing the question of the influence of wheel-running on food preference in connection with sex and meal patterns only with a shorter trial period (2 weeks). Measured parameters are identical between studies with the exception of the Barnes Maze.

Indeed, in my opinion, the current manuscript addresses the research questions/hypotheses stated only vaguely because of the limitations of their selected research design. The former study’s (Yang et al. 2019) work hypotheses were well connected with the design. In this manuscript, however, the authors wanted to investigate the influence of high fat diet on exercise benefits on metabolic and cognitive levels by implementing free diet choice. The design only lets us investigate the response of choice behaviour towards the given diet choices in connection with free access to wheel running or sedentary condition. The variability of diet choice is influenced by time and access to wheel running and individual preferences. Due to the incapability of this design to adjust animals’ diet for a period of time, the capability to find metabolic and cognitive impact of high fat diet, with and without exercise, is highly impaired. The researchers, if they wanted to find the impact of high fat proportion in diet for attenuating exercise effects, should have used fixed diet proportion of fat in the diet on various levels, which would have reduced variability of intake ratio massively. The problem they produced with their design is emphasize in their figure 7; while the sedentary rats have a lower range of diet preference ratio, the exercise group (male) reveal a vast range in ratio, not even overlapping with sedentary. They face even problems with statistic showing large inequality of variance between groups. Authors could only addressed the question of impact of high fat within groups, which is visibly non-existent in the wheel running, nor in the sedentary groups (there is no association between insulin and HF ratio in any within groups analysis – the combination of male sedentary and wheel running groups for the correlation is not possible because of confounding effects of exercise.

A further problem is the integration of measurements for food choices during the last week, where the Barnes Maze test was performed. Indeed, the Barnes Maze is time consuming and will change the environment and experience/challenges for the rats significantly, therefore influencing food choice for certain (as visible from the sudden reduction of energy intake in all groups in the last week (Figure 4) and alterations in diet preference ratio during this week (Figure 3). Authors have no control groups, which would might prove that the Barnes Maze had no impact; therefore the data from the last week’s period need to be excluded from any analysis.

The authors are using very variable terminology for the interpretation of the Barnes Maze. Indeed, the Barnes Maze is certainly an instrument, which can address learning but is influenced by other factors, like fear response, physiological state (i.e. hunger), motivation, speed of locomotion and others. The connection with cognitive dysfunction, the Barnes Maze is not always clear cut and the experiments should have been analyses more carefully. As an example, the sedentary male rats have a higher weight and presumable will be more anxious in the maze – it is well known that exercise reduces anxiety like behaviour in rodents and increases explorative behaviour (Binder et al. 2004). Consequently, the outcome of the manipulations may not be at all connected to any detrimental cognitive effects be the high fat diet in sedentary rats. The data are even in the expected normal range for the Barnes Maze and are not assuming any problems with the task even in reversal (Fedorova et al. 2009). The authors should analyse the data regarding cm distance moved and freezing time– this could explain more about the reasons of the slightly higher latency of the sedentary groups (Figure 8). Consequently, the interpretation and use of terminology for addressing their effects are highly speculative and exaggerated (i.e. terms like cognitive dysfunction, dysregulation, rigidity, and impairment, behavioural flexibility).

Regarding the influence of exercise on the detrimental effects of high fat diet. As mentioned above, the design will not lead to a clear cut information about the influence and interactions of those parameters because of the high and different variability of high fat diet ratio in the groups (Figure 7). The only groups, which could be used for addressing the question, are the female groups, due to similar range of HF diet ratio. Interestingly, in the females, insulin levels are clearly not associated with HF ratio and the sedentary even show lower values – clearly integrating information from Figure 7 and 6 for the females, the females are well protected against the impact of the higher range of High Fat diet – and the metabolic responses are well maintained in normal ranges; an exercise effect would be only visible (or might be) if the female system is challenged beyond what it can compensate. This is even emphasized by the marginal increase of body weight between the females, while this is substantial in the males (Figure 5). Indeed, the better metabolic response of females to high fat diet is well described (as example, Amengual-Cladera et al, 2011; i.e. insulin sensitivity). In addition, the authors should have calculated insulin sensitivity and resistance based on their existent data (see Tran et al. 2003). Consequently, the interpretation of the authors to highlight the reduced response to exercise in female rats is neglecting the fact that the females had no deviations from normal range and therefore a response to exercise was not apparent, which is indeed visible in males – considering the higher insulin levels in sedentary. However, the problem of the huge difference in the choice of High Fat diet in male groups reduces the value of the assumed impact of exercise on metabolism.

Statistics: No sample size calculation was performed – the former work should have given enough opportunity to calculate power and sample for this highly complex design.

Diet analysis: The authors should give choice information also based on mass, not only on calories. It is unlikely that the amount eaten will be a function of calories only but also a matter of mass; the data are biased by the fact that caloric density of the HF food is much higher. Clearly specific satiety will play a role as well, in particular with choice of carbohydrates in connection with exercise (see, Keeley et al. 2014).

Writing style: Introduction is not clear in narrative and uses too many hot terms to increase the study importance, while mixing animal and human studies. Discussion is exaggerating findings and focusing on interpretations of the data, which is not justified in my opinion.

Overall, in my opinion, the study design is not suitable for the set of hypotheses the authors like to address.

Author Response

Reviewer's Comments
Reviewer #1: The manuscript suggests to investigate the influence of high fat diet on the capability of exercise to suppress negative diet effects on body characteristics, metabolism and cognitive function in rats. While the topic is not exhaustively investigated in the literature, I have major concerns about the design, analysis and interpretation of the study.

Major Concerns:
Design

  1. The authors used a free choice paradigm between high fat food and chow (CHO rich) over a period of 6 weeks. The design is identical to their formerly published work (Yang et al. 2019) addressing the question of the influence of wheel-running on food preference in connection with sex and meal patterns only with a shorter trial period (2 weeks). Measured parameters are identical between studies with the exception of the Barnes Maze. Indeed, in my opinion, the current manuscript addresses the research questions/hypotheses stated only vaguely because of the limitations of their selected research design. The former study’s (Yang et al. 2019) work hypotheses were well connected with the design. In this manuscript, however, the authors wanted to investigate the influence of high fat diet on exercise benefits on metabolic and cognitive levels by implementing free diet choice. The design only lets us investigate the response of choice behaviour towards the given diet choices in connection with free access to wheel running or sedentary condition. The variability of diet choice is influenced by time and access to wheel running and individual preferences. Due to the incapability of this design to adjust animals’ diet for a period of time, the capability to find metabolic and cognitive impact of high fat diet, with and without exercise, is highly impaired. The researchers, if they wanted to find the impact of high fat proportion in diet for attenuating exercise effects, should have used fixed diet proportion of fat in the diet on various levels, which would have reduced variability of intake ratio massively.

Authors’ response: The authors appreciate the suggestion that we could take a different approach. Our focus is more on diet choice than a specific effect of different percentages and sources of fat. Fat intake is roughly 30% for the typical Western diet and having a range of fixed dietary proportions of fat may not be as representative to the human condition. Regardless, both designs are complex and would require a multi-group design for better control over the interpretation of the results.

  1. The problem they produced with their design is emphasize in their figure 7; while the sedentary rats have a lower range of diet preference ratio, the exercise group (male) reveal a vast range in ratio, not even overlapping with sedentary. They face even problems with statistic showing large inequality of variance between groups. Authors could only addressed the question of impact of high fat within groups, which is visibly non-existent in the wheel running, nor in the sedentary groups (there is no association between insulin and HF ratio in any within groups analysis – the combination of male sedentary and wheel running groups for the correlation is not possible because of confounding effects of exercise.

Authors’ response: A strength of our current study design is that it also allows us to study individual differences in fat preference that would be difficult using a fixed feeding schedule. The nature of the research question would be better addressed with the inclusion of a naïve chow-fed control and a WR group with only HF diet. We have addressed concerns from point 1 and 2 with a paragraph on the limitations of the current study in the discussion section.

On page 15, lines 499 – 511, the text reads, “Caution should be taken when interpreting the results due to inherent limitations resulting from the complexity of the study design… To our knowledge, no study has assessed changes in insulin sensitivity following HF diet and voluntary exercise in rats of both sexes. The addition of a naïve chow-fed control group is necessary to make between-group comparisons to strengthen the argument for the protective effect of exercise against HF diet. Exercise is more likely to have a greater beneficial effect in males, but the addition of a WR group maintained on only HF diet is necessary to make this conclusion given the shift in HF diet preference across time and the highly variable HF diet intake between subjects and sexes. Nevertheless, our results lend support to our hypotheses regarding the protective effect of exercise against the detrimental outcomes of HF feeding and inform the development of more optimized designs for future studies.”

  1. A further problem is the integration of measurements for food choices during the last week, where the Barnes Maze test was performed. Indeed, the Barnes Maze is time consuming and will change the environment and experience/challenges for the rats significantly, therefore influencing food choice for certain (as visible from the sudden reduction of energy intake in all groups in the last week (Figure 4) and alterations in diet preference ratio during this week (Figure 3). Authors have no control groups, which would might prove that the Barnes Maze had no impact; therefore the data from the last week’s period need to be excluded from any analysis.

Authors’ response: During the last week, all rats went through the same procedures and daily treatment and as a result, total energy intake decreased for all rats. Diet choice patterns, however, were unaffected. More specifically, preference for HF diet changed weekly but during the last week, the diet choice patterns followed the same directional trend as before behavioral testing commenced. Because the diet choice patterns did not shift in the opposite direction, we do not believe that there is sufficient justification to remove the data during behavioral testing from the analyses.

  1. The authors are using very variable terminology for the interpretation of the Barnes Maze. Indeed, the Barnes Maze is certainly an instrument, which can address learning but is influenced by other factors, like fear response, physiological state (i.e. hunger), motivation, speed of locomotion and others. The connection with cognitive dysfunction, the Barnes Maze is not always clear cut and the experiments should have been analyses more carefully. As an example, the sedentary male rats have a higher weight and presumable will be more anxious in the maze – it is well known that exercise reduces anxiety like behaviour in rodents and increases explorative behaviour (Binder et al. 2004). Consequently, the outcome of the manipulations may not be at all connected to any detrimental cognitive effects be the high fat diet in sedentary rats. The data are even in the expected normal range for the Barnes Maze and are not assuming any problems with the task even in reversal (Fedorova et al. 2009). The authors should analyse the data regarding cm distance moved and freezing time– this could explain more about the reasons of the slightly higher latency of the sedentary groups (Figure 8). Consequently, the interpretation and use of terminology for addressing their effects are highly speculative and exaggerated (i.e. terms like cognitive dysfunction, dysregulation, rigidity, and impairment, behavioural flexibility).

Authors’ response: We do not have the ability to assess distance traveled. Distance traveled would not necessarily provide more support as to why sedentary rats had longer escape latencies than their WR counterparts but perhaps give insight into speed/motor control and motivation to escape. Moreover, regardless of the location of the starting quadrant and proximity to the escape box, both sedentary and WR rats turned in the same direction each trial and circled the periphery of the maze to find the escape box. Thus, distance traveled likely did not differ between the sedentary and WR groups.

Based on our observations running and scoring the behaviors, once a rat habituated to the maze, there was no evidence of freezing behavior. All rats immediately started circling the periphery of the maze to escape upon being placed on the maze. Data from one wheel running male rat that froze immediately and repeatedly when placed on the maze was excluded from statistical analyses.

We found little evidence of cognitive “impairment” per se, but there was a difference in cognitive performance among sedentary and WR rats. Cognition is multi-faceted and in the present experiment, we chose to investigate PFC-dependent behavioral flexibility in which the Barnes Maze may not be sensitive enough to unveil subtle deficits. Given the complexity of cognitive behavior and sensitivity of all behavioral tests to extraneous factors, it would be beneficial to assess multiple behaviors using a battery of tests to obtain a clearer picture about how HF diet may affect cognitive function detrimentally. We have revised the language to be less exaggerated based on our findings and included a statement about improvements that can be made during behavioral testing.

On pages 14, lines 491 – 496, the text reads, “…a more sensitive behavioral task may allow us to uncover cognitive deficits more readily than the Barnes maze. Cognition is a complex and multi-faceted construct – while we found evidence for differences in cognitive performance, future studies should include a battery of behavioral tests to tap into different aspects of cognitive behavior that may be adversely influenced by HF diet. Importantly, prolonged HF diet intake and preference may impair specific, rather than global, domains of cognitive function…”

  1. Regarding the influence of exercise on the detrimental effects of high fat diet. As mentioned above, the design will not lead to a clear cut information about the influence and interactions of those parameters because of the high and different variability of high fat diet ratio in the groups (Figure 7). The only groups, which could be used for addressing the question, are the female groups, due to similar range of HF diet ratio. Interestingly, in the females, insulin levels are clearly not associated with HF ratio and the sedentary even show lower values – clearly integrating information from Figure 7 and 6 for the females, the females are well protected against the impact of the higher range of High Fat diet – and the metabolic responses are well maintained in normal ranges; an exercise effect would be only visible (or might be) if the female system is challenged beyond what it can compensate. This is even emphasized by the marginal increase of body weight between the females, while this is substantial in the males (Figure 5). Indeed, the better metabolic response of females to high fat diet is well described (as example, Amengual-Cladera et al, 2011; i.e. insulin sensitivity).

Authors’ response: The reviewer pointed out an important limitation of our current group design, and we deeply appreciate the reference article that the reviewer provided. We agree that the results of male rats are confounded by the fact that male WR rats did not consume comparable amounts of HF diet relative to their sedentary counterparts. Our data pointed out that reduced consumption and preference for HF diet may contribute to the more apparent beneficial metabolic effects of exercise in males than females. However, given that male WR rats did not consume enough HF diet, future studies should include a group of WR with access to HF diet only. We have included this limitation in response to concern #2 (above). Importantly, as described below, both Sed and WR female rats significantly increased their HOMA-IR value to above the 2.60 cutoff indicating that glucose metabolism and insulin sensitivity were worsened by HF feeding in female rats. The Amengual-Cladera article used diets with 2.9% and 26% fat, respectively, as the low and high fat diet. While the findings revealed distinct adaption mechanisms to diet high in fat composition and better metabolic responses in females, they also suggested that the metabolic responses in females may worsen drastically if challenged with a diet much higher in fat composition.

  1. In addition, the authors should have calculated insulin sensitivity and resistance based on their existent data (see Tran et al. 2003). Consequently, the interpretation of the authors to highlight the reduced response to exercise in female rats is neglecting the fact that the females had no deviations from normal range and therefore a response to exercise was not apparent, which is indeed visible in males – considering the higher insulin levels in sedentary. However, the problem of the huge difference in the choice of High Fat diet in male groups reduces the value of the assumed impact of exercise on metabolism.

Authors’ response: Thank you for this excellent advice. We have added two measures of insulin sensitivity – HOMA-IR and ISI0,120 based on the Tran et al. (2003) paper.

On page 4, lines 151 – 154, the text reads, “Blood glucose and plasma insulin data from the baseline and post-HF OGTT was used to assess insulin sensitivity [64] with the following two methods: 1) Hepatic insulin resistance was calculated using the homeostasis model assessment method for insulin resistance (HOMA-IR) model [65] and 2) peripheral insulin resistance was calculated using Gutt’s index of insulin sensitivity (ISI0,120) [66].”

On page 6, lines 211 – 214, the text reads, “Two measures of insulin sensitivity were first calculated using HOMA-IR and ISI0,120 and then analyzed separately using a 2-way mixed model ANOVA with sex and exercise as the between-subjects factors and time (baseline vs. post-HF) as the within-subjects factor.”

On page 10, lines 321 – 332, the text reads, “Male Sed rats had impaired hepatic insulin sensitivity following HF feeding [time x sex x exercise F(1,40) = 7.73, p < 0.01; post hoc M Sed BL vs. post-HF p < 0.0001] as evidenced by increased HOMA-IR index (Table 2). Exercise protected against this detrimental metabolic effect of HF diet in male WR rats by attenuating increases in insulin resistance (post hoc WR BL vs. post-HF p > 0.18, and post-HF Sed vs. WR, p < 0.01). In females, both Sed and WR rats had evidence of insulin resistance after long-term HF diet preference (post hoc Sed and WR BL vs. HF, both p < 0.05) and exercise did not have the same protective effect as seen in males (post-hoc post-HF Sed vs. WR p > 0.08). Peripheral insulin sensitivity was analyzed using ISI0,120 and sex differences in the protective effect of exercise failed to reach statistical significance [time x sex x exercise F(1,43) = 2.29, p > 0.13]. However, a priori t-tests revealed that post-HF feeding, male Sed rats had lower ISI0,120 than their WR counterparts, indicating reduced insulin sensitivity in the Sed but not WR group [t(10,14) = -2.17, p < 0.05]. This difference between Sed and WR groups was absent in females [t(10,13) = -0.21, p > 0.83].”

While exercise did not appear to have a protective effect in females, HF diet did have a detrimental metabolic effect. The cutoff for HOMA-IR indicating signs of insulin resistance is a value ≥ 2.60, and both sedentary and WR females were well above this cutoff (3.15 and 4.63, respectively) in addition to Sed males (5.01). The only group that did not show signs of insulin resistance is the male WR group despite increased preference for HF diet at the end of the experiment. We have incorporated these results and revised our discussion accordingly.

On pages 13 – 14, lines 444 – 454, the text reads, “Our results suggest that that without a concurrent decrease in body weight, adiposity, and HF diet preference, exercise has a limited effect on significantly improving peripheral insulin resistance during chronic access to HF diet [16-18]. Female rats had increased HOMA-IR (Table 2) above the 2.60 cutoff [99] for evidence of insulin resistance after long-term HF feeding regardless of the opportunity to exercise. In contrast, exercise appeared to protect against the development of insulin resistance in males. The results of ISI0,120 also suggest that male WR rats were the only group that maintained insulin sensitivity after six weeks of exposure to HF diet. Although higher HF diet and total energy intake in WR females may play a role in these observed sex differences and additional pair-fed groups will be needed to assess such possibility in future studies, our results reveal that exercise produces more protective effects against insulin resistance in males than females by reducing HF diet preference and consumption.”

Statistics

  1. No sample size calculation was performed – the former work should have given enough opportunity to calculate power and sample for this highly complex design.

Authors’ response: We based the sample size of the present experiment on our previous studies using the same two-diet choice and wheel running paradigm. Previous analyses revealed that the diet choice behavior is robust and the sample size we used was adequately powered to detect a significant difference. More specifically, with an n ≈ 10/group, our observed power (β) is ≥ 0.80, the typical cutoff used for power analyses. In this experiment, each group had at least 10 or more rats, which was sufficient for us to detect group differences in behavior. We have included this information in the Methods.

On page 3, lines 123 – 126, the text reads, “The sample size, n = 10 – 11 for Sed groups and n = 13 – 14 for WR groups, was determined based on our previous studies using the same two-diet choice and wheel running paradigm [61]. The observed power with such sample size was ≥ 0.80, which is the typical cutoff used for power analyses.”

Diet analysis

  1. The authors should give choice information also based on mass, not only on calories. It is unlikely that the amount eaten will be a function of calories only but also a matter of mass; the data are biased by the fact that caloric density of the HF food is much higher. Clearly specific satiety will play a role as well, in particular with choice of carbohydrates in connection with exercise (see, Keeley et al. 2014).

Authors’ response: We agree that both volumetric and nutritive satiety likely play a role in feeding behavior. However, we chose to address the difference in calories between the chow and HF diet by reporting both in kcal. Neither the main statistical results nor diet choice patterns change when reporting feeding patterns based on diet mass (see figures in attached file). Thus, we have kept our original analyses and figures with kcal as the unit of measure.

Writing style

  1. Introduction is not clear in narrative and uses too many hot terms to increase the study importance, while mixing animal and human studies. Discussion is exaggerating findings and focusing on interpretations of the data, which is not justified in my opinion.

Overall, in my opinion, the study design is not suitable for the set of hypotheses the authors like to address.

Authors’ response: When writing, we are highly cognizant of the distinction between preclinical and clinical studies and make sure that we do not overstate and generalize our findings to humans without precaution. We agree that there are certain limitations due to the study design. Appropriate revisions have been made pointing out these limitations and we have also modified the language used to not overgeneralize conclusions. We are transparent about what questions our design is able to answer and have noted improvements that can be made in subsequent studies to more directly address the research question.

To address the reviewer’s concerns, we have made the animal and human studies clear and replaced strong words with ones that are more moderate when describing alterations in metabolic function and behavior in the introduction and discussion (see the manuscript document with “Track Changes”). While our study design can be improved, we believe the data obtained is worthwhile, can inform the development of more optimized study design, and warrants further research.

Reviewer 2 Report

Re the ratings, there is a big difference between Average and high!  I would have liked to have ticked Above average for Overall.  

This is an interesting and potentially important concept that exercise reduced food intake in males but not females.  The authors need to make it clearer f similar findings are shown in humans.    

The rats preferred high fat when wheel running.  Again they need to make it clear if there is evidence that exercise training increases preference for high fat diet in humans?  I doubt it as CHO is more important for energy during exercise of at least moderate intensities than fat.

They need to discuss how wheel running is probably more like a normally active rat in the wild than the sedentary group.  The sedentary group is really non physiologically sedentary.  This is the case in most rodent studies.  Examination of the effects of ex training should have wheel running as a normal rat and adding on treadmill running or equivalent as exercise training.  But it is tricky as that is stressful.  Anyway, these things should be considered in regards to the relevance to humans.

I quite like the paper but it needs to be tightened quite a lot.

Author Response

  1. This is an interesting and potentially important concept that exercise reduced food intake in males but not females. The authors need to make it clearer f similar findings are shown in humans.

Authors’ response: To our knowledge, there is no consensus in the literature regarding the compensatory response to exercise in humans. Although potential sex-dependent changes in food intake following exercise may exist, there have not been sufficient human studies to elucidate this issue. We have added some background information about this in the introduction section.

On pages 1 – 2, lines 42 – 51, the text reads, “Exercise appears to be more effective than diet control at improving metabolic function [16]. People who are successful at maintaining long-term weight loss report high physical activity and a diet low in fat composition [17,18]. Rodent studies show that males consistently decrease food intake in response to exercise [19-22] whereas results are more variable in females [21-23]. In contrast, the majority of human studies focus on changes in food intake following acute rather than long-term exercise and the response to exercise is highly variable in both sexes [24-29]. Although women may loss less weight than men during long-term exercise [30], there has been limited research to elucidate sex differences in exercise-related changes in energy intake and expenditure [31]. Therefore, it is unclear how closely sex differences in energy intake and macronutrient preference during exercise in humans parallel the rodent literature. Furthermore…”

  1. The rats preferred high fat when wheel running. Again they need to make it clear if there is evidence that exercise training increases preference for high fat diet in humans?  I doubt it as CHO is more important for energy during exercise of at least moderate intensities than fat.

Authors’ response: Both male and female WR rats started out avoiding HF diet and gradually increased their intake and preference for HF diet across time with the majority of female rats reversing their initial avoidance earlier than males. Sex differences in macronutrient preference during moderate aerobic exercise in humans has not been directly examined; however, there is evidence that fat is a more efficient fuel source, metabolically speaking, in both sexes and that females preferentially undergo fat oxidation relative to males during exercise. We have included this information in the discussion.

On page 13, lines 416 – 426, the text reads, “…the HF diet is a more efficient fuel source than the standard chow diet, which is higher in carbohydrates than fats [84,85]. Indeed, human studies have shown that both submaximal and endurance exercise results in a higher ratio of lipolysis to carbohydrate oxidation [86-88]. Our previous short-term studies with the same paradigm of wheel running and two-diet choice revealed that the majority of male rats express persistent HF diet avoidance whereas the majority of females reverse HF diet avoidance [76,89-91]. These results are consistent with the report that estradiol enhances lipid metabolism during exercise in rats [92]. When the choice duration was extended, both male and female WR groups preferred the HF diet by the end of the 6-week period (Figure 3E & F). Glucose metabolism is positively correlated with exercise intensity in humans [93] whereas fat oxidation is more likely to occur during low intensity exercise, especially when prolonged [94]. Thus, this shift in fat preference may be a compensatory result of increased energy requirement from long-term aerobic exercise where carbohydrates are no longer the most efficient fuel substrate.”

  1. They need to discuss how wheel running is probably more like a normally active rat in the wild than the sedentary group. The sedentary group is really non physiologically sedentary.  This is the case in most rodent studies.  Examination of the effects of ex training should have wheel running as a normal rat and adding on treadmill running or equivalent as exercise training.  But it is tricky as that is stressful.  Anyway, these things should be considered in regards to the relevance to humans.

Authors’ response: The reviewer brings up an interesting point. Although rats in the wild will voluntarily hop on and run if a wheel is available, we believe that a normally active rat in the wild will be more appropriately compared with a rat housed with other enrichments e.g., nesting materials than with a running wheel. Our sedentary rats were provided with cotton nesting materials. Although these rats may not be as active as those in the wild, we would not consider our rats “non physiologically sedentary”. We have included the enrichment information in the method section.

On page 3, lines 127 – 128, the text reads, “After habituation, sedentary (Sed) rats were moved to standard individual housing cages that included cotton nesting materials as enrichment, while wheel running…”

Furthermore, our focus was specifically on voluntary exercise and diet choice rather than “exercise training”. Nevertheless, having a wider variety of exercise paradigms would lend more support to the effects of exercise on diet preference and energy balance if the results are consistent across different exercise paradigms. We have addressed this as a caveat to interpreting the results from our experiment and how they can be translated to humans.

On page 13, lines 428 – 431, the text reads, “The addition of groups undergoing different types of exercise (e.g., strength, treadmill, swimming, etc.) for different lengths of time would provide additional evidence for the effect of exercise on energy intake and macronutrient preference if the results are consistent.”

On page 15, lines 528 – 531, the text reads, “Here, we focused on voluntary exercise; however, the effect of exercise may shift depending on exercise conditions (forced vs. voluntary, strength vs. endurance, acute vs. chronic, etc.). An extensive investigation is necessary before results can be directly translated to humans.”

  1. I quite like the paper but it needs to be tightened quite a lot.

Authors’ response: Thank you for your review and insightful comments. Together with the responses to other reviewers’ comments, we believe that the quality of the manuscript has improved quite a lot.

Reviewer 3 Report

The study design is well constructed and the aims are clear. However, it would be advantageous from a nutritional aspect to know diets composition (raw materials) and particularly useful the percentage of PUFA and omega 6/3 ratio. Certain these data represent a starting point for further investigations, a future challenge could be done evaluating metabolic and cognitive differences in rats administering an HF diet versus a high protein (HP) diet.

Author Response

  1. The study design is well constructed and the aims are clear. However, it would be advantageous from a nutritional aspect to know diets composition (raw materials) and particularly useful the percentage of PUFA and omega 6/3 ratio. Certain these data represent a starting point for further investigations, a future challenge could be done evaluating metabolic and cognitive differences in rats administering an HF diet versus a high protein (HP) diet.

Authors’ response: We have provided detailed information about the composition of both the chow and HF diet on page 3 in the methods section (Table 1). Neither diet contains omega 6 or 3 as those are typically used to supplement standard diets. Because the chow diet is a purified diet, the macronutrient sources are different from the HF diet and thus, we cannot make any conclusions about the effect of raw materials and the associated cognitive and metabolic outcomes. To address this concern, we currently use an open source control diet matched to the HF diet in our follow up experiments. We have added that improvement to the study design in the discussion section.

On page 15, lines 500 – 503, the text reads, “Different fat sources and composition (e.g., polyunsaturated, monounsaturated) may differentially affect aspects of metabolism and cognition. Future studies should utilize diets matched in macronutrient sources as much as possible to limit the confounding effect of differences in raw materials.”

Given the obesity epidemic, we currently focus on examining the influence of high-fat-high-sugar diets to find insights for treatment approaches to weight management.  However, it would certainly be interesting to compare the effects of a HF vs. high-protein diet.

Reviewer 4 Report

Line 33- Change "is" to "are".

Procedures: Were the male and female rats fed the same amount?  or where they fed differing amounts based upon their body weights? (250-275g (males); 150-175g (females). More detailed description/information is needed. 

Author Response

Line 33- Change "is" to "are".

Authors’ response: The word has been revised accordingly.

Procedures: Were the male and female rats fed the same amount? or where they fed differing amounts based upon their body weights? (250-275g (males); 150-175g (females). More detailed description/information is needed.

Authors’ response: All rats had ad lib access to both the chow and HF diet i.e., the rats choose how much food they want to consume. Food was not provided based on body weight. We have revised the description of our experimental procedures to be more specific.

On page 3, lines 105 – 107, the text reads, “During the experimental period, rats had diet choice between the standard, high carbohydrate chow and a novel 45% HF diet (HF; D12451, Research Diets, New Brunswick, NJ). All groups were fed ad libitum.

Round 2

Reviewer 1 Report

The author's provided excellent feedback to the concerns raised. While I am not entirely convinced that the experimental design is appropriate for the questions addressed, I am aware that reseach is about accuracy of empirical observations  and that interpretations can vary being biased by experiences and opinions. Consequently, I am in favour of publishing the current manuscript in this form.

Author Response

We deeply appreciate that the reviewer spent the time to review our manuscript and provide invaluable feedback. The reviewer's comments will certainly help us design and perform better research in the future.

Reviewer 2 Report

Good that you have increased discussion of the relevance of your findings to humans. 

Typo re “Although women may loss less weight than men…”. Should be lose

I would have thought that there would be more than one paper on this?:  “Although women may loss less weight than men during long-term exercise [30…”.

This depends on the intensity and duration:  “Indeed, human studies have shown that both submaximal and endurance exercise results in a higher ratio of lipolysis to carbohydrate oxidation [86-88]”. Usually during exercise thee RER is above 0.85 so more CHO use than fat use. 

This should say Carbohydrate rather than glucose here:  “Glucose metabolism is positively correlated with exercise intensity in humans [93] whereas fat oxidation…”

There is no doubt that exercise training in humans increases the ability to use fat for energy but is there evidence that endurance training in humans increases fat intake?  Again, I doubt it.

Author Response

Authors’ response: We sincerely thank the editor and reviewer for reviewing our manuscript again and appreciate the opportunity to further improve our manuscript. The point-by-point responses are outlined below. The red texts here are specific changes in the revisions of the resubmitted manuscript, which have been clearly highlighted in the revised manuscript using “Track Changes.”

Reviewer's Comments

Reviewer #4: Good that you have increased discussion of the relevance of your findings to humans.

  1. Typo re “Although women may loss less weight than men…”. Should be lose

Authors’ response: The typo has been corrected.

  1. I would have thought that there would be more than one paper on this?: “Although women may loss less weight than men during long-term exercise [30…”.

Authors’ response: We have added more references [1-3] to support the statement (listed below).

  1. Anderson, J.W.; Konz, E.C.; Frederich, R.C.; Wood, C.L. Long-term weight-loss maintenance: a meta-analysis of US studies. Am J Clin Nutr 2001, 74, 579-584.
  2. Despres, J.P.; Bouchard, C.; Savard, R.; Tremblay, A.; Marcotte, M.; Theriault, G. The effect of a 20-week endurance training program on adipose-tissue morphology and lipolysis in men and women. Metabolism 1984, 33, 235-239, doi:10.1016/0026-0495(84)90043-x.
  3. Williams, R.L.; Wood, L.G.; Collins, C.E.; Callister, R. Effectiveness of weight loss interventions--is there a difference between men and women: a systematic review. Obes Rev 2015, 16, 171-186, doi:10.1111/obr.12241.
  4. This depends on the intensity and duration: “Indeed, human studies have shown that both submaximal and endurance exercise results in a higher ratio of lipolysis to carbohydrate oxidation [86-88]”. Usually during exercise thee RER is above 0.85 so more CHO use than fat use.

Authors’ response: The focus of the present study was on diet preference rather that substrate utilization during exercise. Assessing RER was beyond the scope of the study; however, measuring RER may provide more support to our hypothesis that differences in substrate utilization during exercise may contribute to differences in macronutrient preference. We thank the reviewer for pointing out that RER following exercise normally does not fall below 0.85. We have revised our discussion to reflect the comparison of changes during exercise and differences between sexes.

On page 13, lines 417 – 430, the text reads, “…more efficient fuel source than the standard chow diet, which is higher in carbohydrates than fats [88,89]. Indeed, human studies have shown that there is a crossover effect during which the ratio of lipolysis to carbohydrate oxidation during submaximal and endurance exercise increases [90-92]. While this crossover effect is influenced by exercise duration and intensity, it may partially contribute to differences in macronutrient preference among sedentary and physically active individuals…These results are consistent with the report that estradiol enhances lipid metabolism during exercise in rats [96]. Furthermore, results with respiratory exchange ratio as a measure of substrate utilization from human studies using indirect calorimetry suggest that compared to men, women utilize more fat as the fuel source as a result of long-term exercise [90,91,97,98].  A direct assessment of fuel oxidation will be necessary to support our hypothesis that sex differences in substrate utilization may contribute to differences in running-associated macronutrient preference.”

  1. This should say Carbohydrate rather than glucose here: “Glucose metabolism is positively correlated with exercise intensity in humans [93] whereas fat oxidation…”

Authors’ response: The word “glucose” has been replaced with “carbohydrate.”

  1. There is no doubt that exercise training in humans increases the ability to use fat for energy but is there evidence that endurance training in humans increases fat intake? Again, I doubt it.

Authors’ response: We do not intend to generalize our results to behavior in people. In humans, however, there is a cognitive component in which exercise may motivate individuals to make healthier food choices and decrease fat intake and preference. In contrast, the cognitive component may be less influential on food choices in rodents. We have added a sentence to address this in the discussion section.

On page 13, lines 430 – 434, the text reads, “Additionally, when the choice duration was extended, both male and female WR groups preferred the HF diet by the end of the 6-week period (Figure 3E & F). It is unclear whether this increase in fat preference would occur in humans if the behavior can be examined without the influence of the cognitive component of making healthier food choices in subjects who incorporate regular exercise as a lifestyle.”